# Serial measurement of cytokines strongly predict COVID-19 outcome

**Hasan Selcuk Ozger** [ID][1‡]*, **Resul Karakus** [ID][2‡], **Elif Nazli Kuscu**[1], **Umit Emin Bagriacik**[2], **Nihan Oruklu**[2], **Melek Yaman**[2], **Melda Turkoglu**[3], **Gonca Erbas**[4], **Aysegul Yucel Atak**[2], **Esin Senol**[1]

**1** Department of Infectious Diseases and Clinical Microbiology, Faculty of Medicine, Gazi University, Ankara, Turkey, **2** Department of Immunology, Faculty of Medicine, Gazi University, Ankara, Turkey, **3** Department of Internal Medicine and Intensive Care, Faculty of Medicine, Gazi University, Ankara, Turkey, **4** Department of Radiology, Faculty of Medicine, Gazi University, Ankara, Turkey

‡ These authors contributed equally to this work, and all should be considered first authors.
* sozger@yahoo.com

## Abstract

### Purpose

Cytokines are major mediators of COVID-19 pathogenesis and several of them are already being regarded as predictive markers for the clinical course and outcome of COVID-19 cases. A major pitfall of many COVID-19 cytokine studies is the lack of a benchmark sampling timing. Since cytokines and their relative change during an infectious disease course is quite dynamic, we evaluated the predictive value of serially measured cytokines for COVID-19 cases.

### Methods

In this single-center, prospective study, a broad spectrum of cytokines were determined by multiplex ELISA assay in samples collected at admission and at the third day of hospitalization. Appropriateness of cytokine levels in predicting mortality were assessed by receiver-operating characteristic (ROC) analyses for both sampling times in paralel to conventional biomarkers.

### Results

At both sampling points, higher levels of IL-6, IL-7, IL-10, IL-15, IL-27 IP-10, MCP-1, and GCSF were found to be more predictive for mortality (p<0.05). Some of these cytokines, such as IL-6, IL-10, IL-7 and GCSF, had higher sensitivity and specificity in predicting mortality. AUC values of IL-6, IL-10, IL-7 and GCSF were 0.85 (0.65 to 0.92), 0.88 (0.73 to 0.96), 0.80 (0.63 to 0.91) and 0.86 (0.70 to 0.95), respectively at hospital admission. Compared to hospital admission, on the 3rd day of hospitalization serum levels of IL-6 and, IL-10 decreased significantly in the survivor group, unlike the non-survivor group (IL-6, p = 0.015, and IL-10, p = 0.016).

**Data Availability Statement:** All relevant data are within the paper and its Supporting information files.

**Funding:** The funders had no role in study design, data collection and analysis, decision to publish, or preparation of the manuscript.

**Competing interests:** The authors have declared that no competing interests exist.

## Conclusion

Our study results suggest that single-sample-based cytokine analyzes can be misleading and that cytokine levels measured serially at different sampling times provide a more precise and accurate estimate for the outcome of COVID-19 patients.

## Introduction

On December 31, 2019, the World Health Organization (WHO) China Country Office reported cases of pneumonia of unknown etiology in Wuhan, China's Hubei province [1]. On January 7, 2020, the agent was identified as a new coronavirus (2019-nCoV) not previously detected in humans [1]. Later, this novel clinical entity was named as COVID-19, and the virus was named Severe Acute Respiratory Syndrome Coronavirus-2 (SARS-CoV-2) due to its close similarity to SARS CoV [1]. As of October 2021, the COVID-19 outbreak is on the rise, affecting about 243 million people and causing 4.9 million deaths worldwide [2]. Clinical spectrum of COVID-19 vary drastically among patients. COVID-19 disease may progress with completely asymptomatic or mild symptoms, as well as causing lower respiratory tract infections and pneumonia that tend to be self-limiting [1, 3]. In a rare group of patients, it may progress more severely, causing rapidly progressive severe pneumonia, and may result in Acute Respiratory Distress Syndrome (ARDS), multiple organ failure, and death [1, 3].

SARS-CoV-2 infections reveal a unique and inappropriate inflammatory response in patients with worse outcome [3]. This response is characterized by low levels of type I interferons, juxtaposed to elevated chemokines and high expression of some cytokines [3]. Aberrant inflammation observed in COVID-19 course is similar to that observed with other viral respiratory infections including H5N1 influenza, SARS-CoV, Middle East Respiratory Syndrome *Coronavirus* (MERS-CoV) [4]. This hyper-inflammatory state called as 'Cytokine Storm', is consistently associated with more severe disease and worse outcome in COVID-19 [1, 5, 6]. The effect of serum cytokine profiles on the severity of COVID-19 patients has been evaluated in numerous studies [5–8]. In a meta-analysis, serum levels of interleukin-2 (IL-2), IL-2R, IL-4, IL-6, IL-8, IL-10, tumor necrosis factor-alpha (TNF-α), and interferon-gamma (INF-γ) were found to be associated with severe COVID-19 cases [7].

Among these studies, there appear to be differences in single cytokines associated with clinical severity, which is thought to be due to the definition of severe and critical COVID-19 cases and the heterogeneity of the sampling time of sera for cytokine analyses [1, 6, 9]. Although many studies regarding the role of cytokines in COVID-19 pathogenesis have already been published, there are none, to our knowledge, comparing serum cytokine levels at different time points. Therefore, the hypothesis of our study was based on the change in cytokine levels and the effect of this change on COVID-19 severity. Our study aims to investigate the value of change in cytokine levels in predicting COVID-19 mortality by comparing serum cytokine levels at different sampling times.

## Material and methods

### Study design

This single-center, prospective study involved hospitalized COVID-19 patients at the Gazi University Hospital, Ankara, Turkey between June and August 2020. It was approved by the

Gazi University Clinical Studies Ethical Committee (Decision date: 08.06.2020, Decision number: 384). Written informed consent was obtained from all participants.

## Study population, groups, and definitions

The diagnosis of COVID-19 infection was confirmed by polymerase chain reaction (PCR) from nasopharyngeal and oropharyngeal samples in all hospitalized COVID-19 patients enrolled in the study. Patients under 18 years old and patients who received immunosuppressive or immunomodulatory therapy (corticosteroids; IL-6, IL-1 antagonists) were excluded.

## Study protocol

During the study period, all eligible participants were enrolled consecutively. On admission demographical-clinical characteristics, radiological findings, routine laboratory results including hemogram, liver and renal function tests, D-dimer, ferritin, C-reactive protein (CRP), and procalcitonin, and outcomes (28th-day mortality) of patients were recorded.

In addition to routine tests, 3 ml of blood samples were taken at admission and at the $3^{rd}$ day of hospitalization. Serum was separated from whole blood by centrifugation (Allegra® X-12R, Beckman Coulter, USA) at 2500 rpm for 15 minutes and sera were kept at -80˚C until studied.

## Measurement of cytokines

Serum cytokine and chemokine levels were determined by a multiplex ELISA Assay kit, MILLIPLEX® Human Cytokine/Chemokine/Growth Factor Panel A, 48Plex (Merck Millipore, USA).

The cytokines were measured and grouped into four categories according to their function: interleukins (IL-1α, IL-1β, IL-1RA, IL-2, IL-3, IL-4, IL-5, IL-6, IL-9, IL-10, IL-12p40, IL-12p70, IL-13, IL-15, IL-17A, IL-17F, IL-18, IL-22, IL-25 (formerly IL-17E)), chemokines (Eotaxin (CCL11), Fractalkine (CX3CL1), GROα (CXCL1), IL8 (CXCL8), IP-10 (CXCL10, IFNγ-inducible protein 10), MCP-1 (CCL2, monocyte chemotactic protein), MCP-3 (CCL7), MDC (CCL22, macrophage derived chemokine), MIG (CXCL9; Monokine induced by γIFN), MIP-1α (CCL3; Macrophage inflammatory protein 1α), MIP1β (CCL4)), RANTES (= CCL5), cytokines involed in cellular growth and/or development of the immune system, including members of the TNF super familiy (EGF, FGF-2, IL-7, FLT-3L, GCSF, GM-CSF, MCSF, PDGF-AA, PDGF-AA/BB; TGFβ, TGFα; TNFα, TNFβ, sCD40L;), and interferons (IFN-α2, IFN-γ). The assay was conducted according to the manufacturer's recommendations. Briefly, 20 microliters of a serum were loaded into the wells of the test plate. After the final reaction, the plate was read and analyzed using a Luminex xMAP system (Luminex xMAP®, Merck Millipore, USA), interfaced with Luminex XPONENT 4.2 and MILLIPLEX® Analyst 5.1 software.

TGF-β was measured using a sandwich ELISA (Elabioscience, USA). 50 microliters of serum per sample were used in each well of the ELISA plate. Serum samples were run according to the kit instructions. Reactions were read at 450 nm using a plate reader (Synergy HT, BioTek, USA).

## Statistical analysis

All data were analyzed by IBM SPSS Statistics, version 20.0 (IBM Corp., Armonk, N.Y., USA). The normality of the data distribution was determined by the Shapiro-Wilk test, histogram, and Q-Q plots. The categorical values were expressed as numbers and a percentages and were

analyzed with a Chi-square test. Continuous values were presented as a mean and standard deviation (SD) or median values and an interquartile range (IQR) of 25%–75%. The non-parametric values were analyzed using the Mann–Whitney U, and the parametric ones with a Student t-test. Comparison of serum levels of cytokines between baseline and 3rd days of hospitalization were made by paired Student t-test for normal distribution variables and Wilcoxon test for variables that did not show normal distribution. To evaluate the utility of various cytokines and some biomarkers at varying cut-off values for COVID-19-associated mortality, a receiver-operating characteristic (ROC) curve was generated, and the area under the curve (AUC) was calculated. The ROC curves obtained for predicting COVID-19-related mortality for some cytokines were compared. MdCalc version 20.009 program was used to compare ROC curves. The 95% confidence intervals (95% CIs) were also calculated when appropriate, and a p-value <0.05 was considered statistically significant.

## Results

37 COVID-19 patients were enrolled in the study. The median age of COVID-19 patients was 61 (IQR 25–75%: 50 to 72) and 24 (64.9%) of them were male. The median symptom duration of patients was 2 days (0 to 3.5 days). Eight (21.6%) of 37 patients had a mortal outcome within 28 days. The baseline characteristics grouped according to the outcome (Table 1).

The cytokine levels of the non-survivor and survivor groups at admission were analyzed and were presented in Table 2.

The differences of some cytokine levels at admission and the 3rd day of hospitalization were also evaluated in survivor and non-survivor groups. While IL-6 and IL-10 levels remained stable in the non-survivor group,(p = 0.208, and p = 0.183, respectively), serum levels of IL-6, IL-10 were significantly decreased over the 72-hour period in the survivor group, (Wilcoxon test, p = 0.015, and 0.016, respectively (Fig 1).

To evaluate the utility of various cytokines and some biomarkers at varying cut-off values for the 28th day of mortality, a ROC curve was generated, and the area under the curve (AUC) was calculated. The AUC, PLR, and NLR of cytokines and biomarkers for predicting mortality were presented in Table 3 and Fig 2.

## Discussion

In our study, it was determined that IL-6, IL-7, IL-10, IL-15, IL-27, IP-10, MCP-1, and GCSF cytokines on day of admission could be used for prediction of mortality in COVID-19 patients. At admission, it was found that some of these cytokines (IL-15, IP-10, IL-27, MCP-1) had limited contribution in predicting mortality and did not have significant advantages compared to biomarkers such as CRP, D-dimer, and ferritin. However, some of these, particularly IL-6, IL-10, IL-7 and GCSF, had higher sensitivity and specificity in predicting mortality And also, the sensitivity and specificity of cytokines such as IL-6, IL-10, IL-15 and MCP-1 for predicting mortality increased with prospective follow-up. Compared to hospital admission, on the 3rd day of hospitalization serum levels of IL-6 and, IL-10 decreased significantly in the survivor group, unlike the non-survivor group. These dynamic changes in cytokine levels indicate that serial measurement of certain cytokine levels might be useful to predict outcomes in COVID-19 patients. Observed decreases in those aferomentioned cytokines in the survivor cases could also be regarded as good prognostic markers for COVID-19 outcome. However, changes in cytokine levels at different time points did not add any additional contribution to prognosis prediction when compared with changes in prognostic markers such as CRP or D-dimer.

IL-6 is regarded as one of the major pro-inflammatory interleukins and involved in many immunological processes including induction of acute phase responses [10, 11]. The effects of

**Table 1. Demographical and clinical characteristics of patients according to mortality.**

| Variables | All Patients n: (37) | Non-survivors (n: 8) | Survivors (n: 29) | p-value |
|---|---|---|---|---|
| Gender *n (%)* | | | | |
| Male | 24 (64.9) | 4 (50) | 20 (69) | 0.413 |
| Age *median (IQR 25–75%)* | 61 (50 to 72) | 73 (62.8 to 82.7) | 61 (40.5 to 70) | 0.009 |
| Presence of Comorbidities n *(%)* | | | | |
| Chronic hypertension | 20(54.1) | 7 (87.5) | 13 (44.8) | 0.048 |
| Diabetes mellitus | 10 (27) | 3 (37.5) | 7 (24.1) | 0.655 |
| Cardiovascular disease | 6 (16.2) | 1 (12.5) | 5 (17.2) | 1.000 |
| Chronic kidney disease | 4 (10.8) | 1 (12.5) | 3 (10.3) | 1.000 |
| Malignancy | 8 (21.6) | 2 (25.0) | 6 (25.7) | 1.000 |
| COPD | 3 (8.1) | 0 | 3 (10.3) | N/A |
| Asthma | 5 (13.5) | 2 (25.0) | 3 (10.3) | 0.292 |
| Patients with at least one comorbidity *n(%)* | 28 (75.7) | 8 (100) | 20 (69) | 0.159 |
| Patients with at least two comorbidities *n(%)* | 17 (45.9) | 6 (75) | 11 (37.9) | 0.109 |
| ACE inhibitors and ARB | 11 (29.7) | 4 (50) | 7 (24.1) | 0.203 |
| Smoking history *n (%)* | 7 (18.9) | 1 (12.5) | 6 (20.7) | 1.000 |
| Symptoms on admission *n (%)* | | | | |
| Fever>38˚C | 13 (35.1) | 2 (25) | 11(37.9) | 0.685 |
| Cough | 13 (35.1) | 2 (25) | 11(37.9) | 0.685 |
| Dyspnea | 17(45.9) | 6 (75) | 11 (37.9) | 0.109 |
| Sputum | 8 (21.6) | 2 (25) | 6 (20.7) | 1.000 |
| Laboratory findings on admission *median (IQR 25–75%)* | | | | |
| White blood cell, $mm^3$, mean ± SD | 8.50 ±3.5 | 10.2 ± 4.5 | 8.01 ± 3.01 | 0.089 |
| Leukocyte, $mm^3$ | 5.41 (3.19 to 8.58) | 9.07 (3.19 to 12.3) | 4.9 (3.1 to 7.3) | 0.251 |
| Lymphocyte, $mm^3$ | 1.25 (0.96 to 2.03) | 1.18 (0.64 to 2.41) | 1.25 (0.96 to 2.01) | 0.786 |
| Hemoglobin, g/dL, mean ± SD | 11.9 ± 2.9 | 11.03 ± 2.8 | 12.1 ± 2.9 | 0.935 |
| Platelet, x $10^3/mm^3$ | 238 (194 to 232) | 256 (213 to 335) | 235 (186 to 267) | 0.317 |
| Creatinine, mg/dL | 0.95 (0.74 to 1.28) | 1.1 (0.64 to 1.91) | 0.94 (0.74 to 1.14) | 0.704 |
| AST, U/L | 25 (21.5 to 38) | 29.5 (21 to 81) | 25 (21.5 to 34) | 0.221 |
| LDH, U/L | 249 (198 to 321) | 385 (285 to 762) | 222 (186 to 297) | 0.003 |
| INR | 1.11 (0.99 to 1.29) | 1.19 (1.0 to 1.33) | 1.0 (0.99 to 1.29) | 0.625 |
| Troponin, ng/L | 10 (5 to 65) | 68 (19 to 95) | 8 (5 to 19) | 0.015 |
| Lactate | 1.6 (1.4 to 1.9) | 1.4 (1.1 to 3.8) | 1.6 (1.3 to 2.07) | 0.720 |
| D-dimer, µg/mL | 1.0 (0.37 to 2.16) | 2.43 (1.16 to 4.69) | 0.67 (0.31 to 1.61) | 0.009 |
| Ferritin, ng/mL | 197 (37 to 735) | 536 (242 to 1894) | 105 (32 to 531) | 0.046 |
| Fibrinogen, mg/dL, mean ± SD | 478 ± 175 | 621 ± 125 | 436 ± 136 | 0.418 |
| CRP, mg/dL | 52.6 (12.5 to 99.3) | 92 (55 to 162) | 36 (8.8 to 89.4) | 0.056 |
| Procalcitonin, ng/mL | 0.08 (0.03 to 0.15) | 0.54 (0.22 to 1.56) | 0.08 (0.03 to 0.15) | 0.02 |

*IQR*, Inter-quartile range; *COPD*, Chronic obstructive pulmonary disease; *ACE*, Angiotensin-converting enzyme; *NSAI*, non-steroidal anti-inflammatory; *CRP*, C-reactive protein; *ICU*, Intensive care unite; N/A, Not applicable

IL-15 are also pro-inflammatory [11]. IL-10 on the other hand, by a direct effect on macrophages and T and B-cells has a major immune suppressive function and may counter-act to control this (hyper)inflammatory state caused by compound action of pro-inflammatory cytokines [12, 13]. The association of these cytokines with COVID-19 severity has also been described in some previous studies [1, 9, 14–22]. In our study, these cytokines were also found to be high in the non-survivor group. Also, on the 3rd day of hospitalization, there was a

**Table 2. The cytokine levels of COVID-19 patients according to mortality.**

| | Hospital admission | | |
|---|---|---|---|
| | **Non-survivors (n: 8)** | **Survivors (n:29)** | **p value** |
| Interleukins, median, IQR (25 to 75%) | | | |
| IL-1α | 3.2 (2.3 to 4.6) | 2.51 (2.31 to 8.57) | 0.928 |
| IL-1β | 3.3 (1.8 to 9.6) | 2.3 (1.6 to 6.6) | 0.651 |
| IL-1RA | 48 (7.1 to 90.1) | 7.6 (3.7 to 14.8) | 0.094 |
| IL-2 ¶ | - | - | NA |
| IL-3 ¶ | - | - | NA |
| IL-4 | 0.6 (0.6 to 1.2) | 0.8 (0.6 to 1.5) | 0.871 |
| IL-5 | 5.2 (2.8 to 76) | 3.3 (2.0 to 6.9) | 0.251 |
| IL-6 | 161 (23 to 1283) | 15 (5.7 to 38) | 0.005 |
| IL-7 | 11.8 (10 to 17.8) | 5.9 (2.3 to 9.8) | 0.008 |
| IL-9 | 18.7 (13 to 23) | 19.1 (13 to 24) | 0.695 |
| IL-10 | 47 (31 to 185) | 11 (2.3 to 26.7) | <0.001 |
| IL-12p40 | 40 (27 to 95) | 40 (26 to 88) | 0.986 |
| IL-12p70 | 2.2 (1.4 to 4.1) | 2.4 (1.4 to 3.9) | 0.955 |
| IL-13 | 11.8 (8.5 to 25) | 11.9 (8.5 to 38) | 0.526 |
| IL—15 | 22 (14 to 27) | 9 (6.7 to 11.9) | 0.009 |
| IL-17A ¶ | - | - | NA |
| IL-17F ¶ | - | - | NA |
| IL-18 | 85 (37 to 423) | 51 (22 to 90) | 0.137 |
| IL-25 | 167 (99 to 242) | 181 (107 to 384) | 0.574 |
| IL-22 ¶ | - | - | NA |
| IL-27 | 3477 (2919 to 4318) | 1895 (1031 to 3013) | 0.029 |
| Chemokines, median, IQR (25 to 75%) | | | |
| Eotaxin, mean ± SD | 130 ± 40.6 | 134 ± 57 | 0.410 |
| Fractalkine | 124 (53 to 160) | 102 (67 to 232) | 1.000 |
| GRO-α | 40.1 (29.1 to 60.1) | 35.4 (3 to 62) | 0.731 |
| IL-8 | 56.6 (17.8 to 108) | 33 (15 to 139) | 0.675 |
| IP-10 | 610 (174 to 5594) | 114 (49 to 494) | 0.021 |
| MCP-1 | 1350 (765 to 2297) | 581 (420 to 825) | 0.015 |
| MCP-3 | 28 (11 to 37) | 16 (11 to 59) | 0.780 |
| MDC | 402 (321 to 617) | 621 (358 to 748) | 0.373 |
| MIG | 4034 (2535 to 7728) | 1829 (1030 to 4222) | 0.073 |
| MIP-1α | 23 (16 to 43) | 21 (12 to 37) | 0.335 |
| MIP-1β | 41 (17 to 63) | 35 (23 to 50) | 0.704 |
| RANTES | - | - | NA |
| Cytokines involved in cellular growth and/or development of the immune system, including members of the TNF superfamily, median, IQR (25 to 75%) | | | |
| EGF | 29.1 (6.4 to 133) | 75.8 (45.3 to 135.2) | 0.266 |
| FGF-2 | 86.1 (44.3 to 101) | 52 (38 to 83) | 0.299 |
| FLT-3L | 34 (20 to 42) | 26 (16.7 to 32.2) | 0.266 |
| GCSF | 91 (59 to 200) | 22 (8.3 to 42) | 0.001 |
| GM-CSF ¶ | - | - | NA |
| MCSF | 768 (581 to 775) | 169 (122 to 819) | 0.073 |
| sCD40L | 5676 (778 to 20707) | 7665 (4128 to 11832) | 0.688 |
| PDGF-AA, mean ± SD | 5011 ± 2884 | 4283 ± 1979 | 0.656 |
| PDGF-AB/BB, mean ± SD | 20648 ± 8175 | 20886 ± 7669 | 0.639 |
| TGF-α | 7.4 (3.0 to 19.1) | 5.5 (2.6 to 11.9) | 0.599 |

*(Continued)*

**Table 2.** (Continued)

| | Hospital admission | | |
| --- | --- | --- | --- |
| | **Non-survivors (n: 8)** | **Survivors (n:29)** | **p value** |
| TGF-β | 3.50 (2.74 to 4.73) | 4.05 (2.69 to 5.06) | 0.550 |
| TNF-α | 59.6 (45 to 69) | 39.2 (21 to 69) | 0.156 |
| TNF-β | 2.7 (0.91 to 11.1) | 2.8 (0.87 to 7.91) | 0.957 |
| VEGFA | 362 (153 to 658) | 238 (111 to 465) | 0.166 |
| Interferons, median, IQR (25 to 75%) | | | |
| IFNa2 | 15.9 (9 to 19.9) | 12.8 (8.63 to 26.4) | 0.985 |
| IFN-gamma | 2.08 (1.43 to 4.56) | 1.94 (1.43 to 9.34) | 0.379 |

¶ Results obtained for IL-2, IL-3, IL-17A, IL-17F, IL-22, RANTES, and GM-CSF were at or below the lower detection limit of the immunoassay system for most of the cases, which made it impossible to perform a comparison between those levels, therefore these cytokines were no further evaluated and not included in association analyses

significant decrease in IL-6 and IL-10 cytokine levels in the non-survivor group, in contrast to the high and stable serum levels in the non-survivor group. This different and dynamic change in IL-6 and IL-10 levels, especially the decrease in the serum levels in the survivor group, has led to the clarification of the predictive role of cytokines on the prognosis of COVID-19 with prospective follow-up.

A similar dynamic change was observed in levels of chemokine ligands that act as «*chemo*attractant cyto*kines*». These molecules direct traffic between circulating cells and their migration patterns into peripheral tissues both at homeostasis and under conditions characterized by inflammation [3]. Increases in levels of IP-10, MCP-1, IL-8, MCP-3, and MIP-1a were found to be associated with the severity of COVID-19 [1, 8, 9, 23]. Similarly, in our study, levels of IP-10 and MCP-1 at admission were found to be significantly elevated in the non-survivor group. Compared to survivor group, higher serum levels of IP-10 and MCP-1 over 72 hours in the non-survivor group, possibly denoting an ongoing monocytic efflux from the circulation. In our study, most of the GFs and CSFs were also evaluated. IL-7 which is especially important in T cell differentiation and GCSF denoting an ongoing cellular mobilization were higher in the non-survivor groups [24]. Although a significant difference was not detected in the survivor and non-survivor groups in serum levels at different sampling timefor these cytokines, the roles of these cytokines in predicting mortality varied at different sampling times. Most of the cytokine studies related with COVID-19 do not even state the sampling time, which is an important reference for a dynamic, changing process such as the levels observed in sera. These results suggest that single assessment-based analyses may be less predictive and studies based on a single assessment with an uncertain sampling time may contain misleading results.

There were several limitations in our study. First, it was a single-center and small sample study of patients admitted to the hospital, a larger cohort would be better to assess the temporal change of immune response after infection with COVID-19. Second, the measurement time of cytokine levels was determined by taking into account the hospital admission. This increases the possibility of heterogeneity of the disease duration and associated immune response in the patients enrolled in our study. Third, some conditions, such as age, which could affect the results of the immune response between groups, were not adjustable because of small sample size of the study. Fourth, the regression analyzes that can be performed to determine the role of cytokines in mortality independent of the effect of confounders, such as comorbidities, age etc., cannot be performed due to the current sample size, especially in the

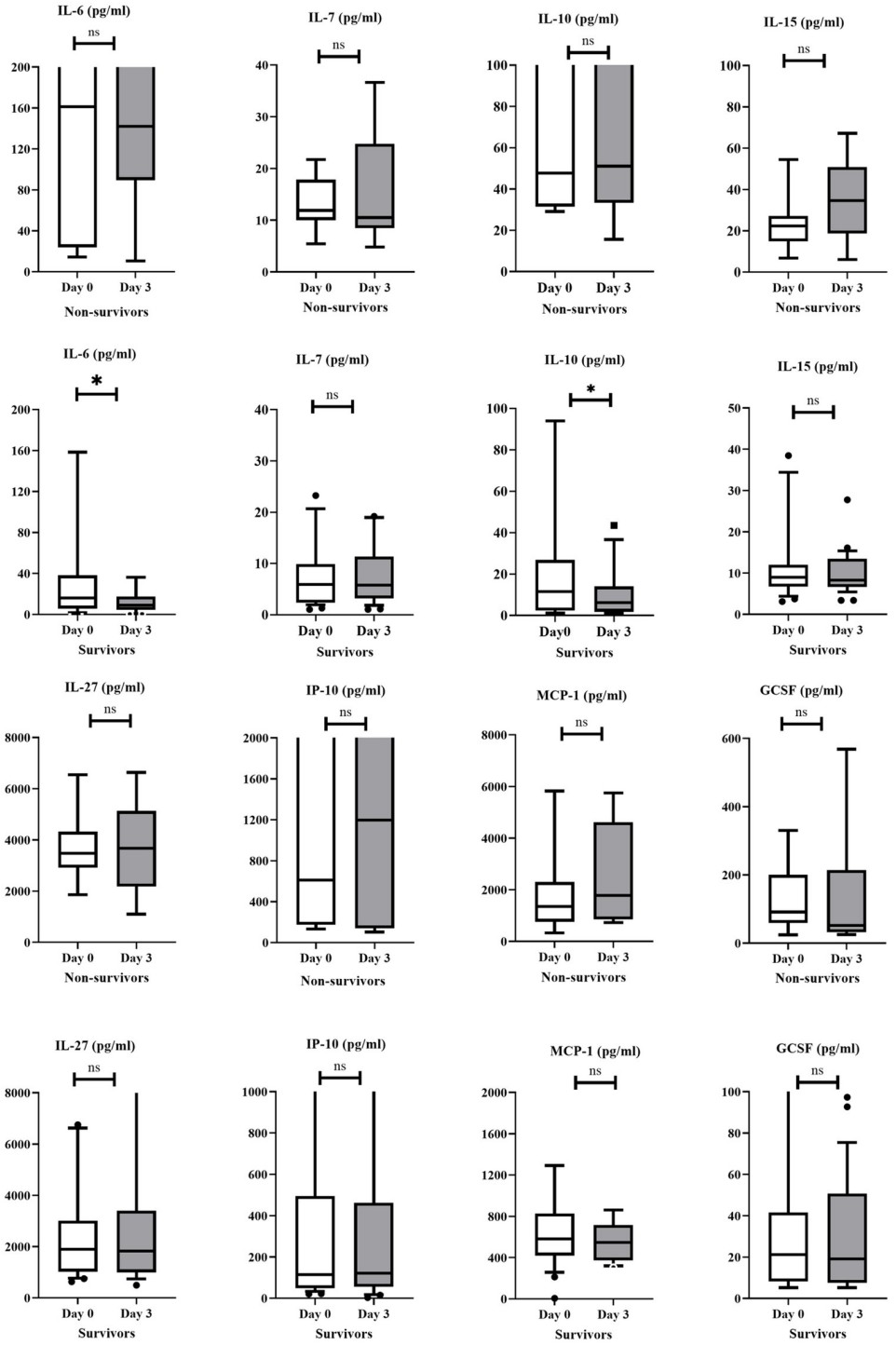

**Fig 1. Evaluation of changes in some cytokine levels in survivor and non-survivor groups.**

non-survivor group. Fifth, results for some cytokines were at or below the lower detection limit of the immunoassay system for most of the cases, which made it impossible to perform a comparison between those levels, therefore these cytokines were no further evaluated and not included in association analyses. However, the most powerful aspect of our study is the

**Table 3. Values of changes in cytokine and some biomarkers levels for prediction of mortality in COVID-19 cases.**

| | Day 0 | | | | Day 3 | | | | Δ | | | |
|---|---|---|---|---|---|---|---|---|---|---|---|---|
| | AUC 95% CI | Optimal cut-off value | PLR 95% CI | NLR 95% CI | AUC 95% CI | Optimal cut-off value | PLR 95% CI | NLR 95% CI | AUC 95% CI | Optimal cut-off value | PLR 95% CI | NLR 95% CI |
| GCSF | 0.86 (0.70–0.95) | 43.7 | 4.2 (2.0–9.0) | 0.16 (0.02–1.0) | 0.78 (0.61–0.90) | 23.1 | 2.2 (1.5–3.3) | 0.23 (0.04–1.5) | 0.58 (0.40–0.74) | 45.6 | 4.83 (1.3–17.3) | 0.56 (0.3–1.1) |
| IL-6 | 0.81 (0.65–0.92) | 48.8 | 4.35 (1.8–10.6) | 0.3 (0.09–1.0) | 0.92 (0.78–0.98) | 53.7 | 25.3 (3.6–177) | 0.13 (0.02–0.8) | 0.69 (0.51–0.83) | -86.3 | 18.1 (2.5–133) | 0.39 (0.2–1.0) |
| IL-7 | 0.80 (0.63–0.91) | 9.39 | 3.6 (1.8–7.3) | 0.16 (0.03–1.0) | 0.73 (0.56–0.86) | 7.91 | 2.3 (1.4–3.9) | 0.2 (0.03–1.3) | 0.51 (0.34–0.68) | 1.61 | 2.07 (0.8–5.3) | 0.66 (0.3–1.4) |
| IL-10 | 0.88 (0.73–0.96) | 24.4 | 4.1 (2.1–7.9) | 0.16 (0.02–1.0) | 0.93 (0.80–0.99) | 15.0 | 5.8 (2.6–12.9) | 0.15 (0.02–1.0) | 0.64 (0.46–0.79) | 50.5 | 10.8 (1.3–90.9) | 0.67 (0.4–1.2) |
| IL-15 | 0.79 (0.63–0.91) | 12.3 | 4.2 (2.0–9.0) | 0.16 (0.02–1.0) | 0.88 (0.73–0.96) | 16.1 | 25.3 (3.6–177) | 0.13 (0.02–0.8) | 0.74 (0.57–0.87) | -12.7 | 14.5 (1.9–112) | 0.5 (0.3–1.0) |
| IL-27 | 0.75 (0.58–0.88) | 2626 | 3.6 (1.8–7.3) | 0.16 (0.03–1.0) | 0.72 (0.54–0.85) | 2649 | 2.72 (1.3–5.5) | 0.35 (0.1–1.2) | 0.53 (0.35–0.69) | 735 | 07 (0.4–1.2) | 3.62 (0.9–14.6) |
| IP-10 | 0.76 (0.60–0.89) | 114 | 2.0 (1.4–3.0) | 0.24 (0.04–1.6) | 0.78 (0.61–0.89) | 915 | 6.04 (1.8–20) | 0.4 (0.2–1.0) | 0.69 (0.51–0.83) | 109 | 1.8 (1.1–2.9) | 0.24 (0.04–1.7) |
| MCP-1 | 0.78 (0.61–0.89) | 966 | 4.35 (1.8–10.) | 0.3 (0.09–1.0) | 0.94 (0.82–0.99) | 722 | 5.8 (2.6–12.9) | 0.15 (0.02–1.0) | 0.53 (0.36–0.70) | -1022 | 10.8 (1.3–90.9) | 0.65 (04–1.1) |
| D-dimer | 0.79 (0.63–0.91) | 2.2 | 6.0 (1.8–20) | 0.4 (0.2–1.0) | 0.89 (0.74–0.96) | 2.3 | 12.6 (3.2–49.6) | 0.13 (0.02–0.8) | 0.81 (0.65–0.92) | -0.8 | 10.8 (2.7–43.9) | 0.27 (0.08–0.9) |
| Ferritin | 0.76 (0.57–0.87) | 229 | 2.82 (1.5–5.2) | 0.18 (0.03–1.2) | 0.81 (0.64–0.92) | 809 | 4.2 (2.0–9.0) | 0.16 (0.02–1.0) | 0.55 (0.38–0.71) | -764 | 7.2 (1.6–32) | 0.5 (0.3–1.1) |
| CRP | 0.72 (0.55–0.85) | 40.6 | 2.3 (1.6–3.5) | 0.21 (0.03–1.4) | 0.86 (0.71–0.95) | 78 | 4.83 (2.4–9.9) | 0.16 (0.02–1.0) | 0.82 (0.66–0.93) | -32.7 | 6.3 (2.5–16.4) | 0.15 (0.02–0.9) |

CRP, C-reactive protein; AUC, Area under the curve; PLR, Positive likelihood ratio; NLR, Negative likelihood ratio

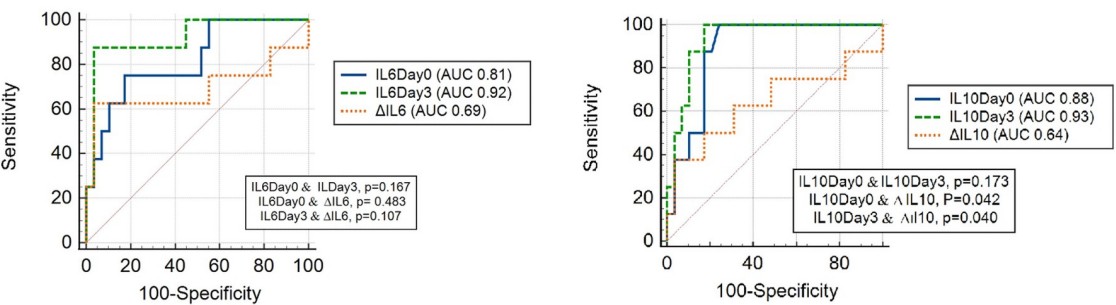

**Fig 2. Comparison of ROC curves in predicting COVID-19-related mortality for serum IL-6 and IL-10 levels.**

determination of cytokine levels at different time points for same patients and the relationship between the dynamic change in cytokine levels and mortality.

## Conclusions

Our study results suggest that IL-6, IL-7, IL-10, IL-15, IL-27, IP-10, MCP-1, and GCSF might be used to predict mortality in COVID-19 patients. According to our results, serial sampling of some cytokines seem to be more predictive for the outcome in comparison to a single measurement, as frequently reported even without a definite sampling time. However, since the measurement of cytokines/chemokines is time-consuming and expensive, serial measurements can be made only for certain cytokines such as IL-6 and IL-10. Possibly, kinetic studies with higher numbers of subjects and more frequent serial measurements of those cytokines found to be important in our paper might yield useful markers for the follow-up and outcome of COVID-19 patients.

## Supporting information

**S1 Data.**
(SAV)

## Acknowledgments

The funders had no role in study design, data collection and analysis, decision to publish, or preparation of the manuscript.

## Author Contributions

**Conceptualization:** Resul Karakus, Umit Emin Bagriacik, Aysegul Yucel Atak, Esin Senol.

**Data curation:** Resul Karakus, Elif Nazli Kuscu, Umit Emin Bagriacik, Aysegul Yucel Atak, Esin Senol.

**Formal analysis:** Hasan Selcuk Ozger, Elif Nazli Kuscu, Nihan Oruklu, Melek Yaman.

**Investigation:** Hasan Selcuk Ozger, Resul Karakus, Nihan Oruklu, Melek Yaman.

**Methodology:** Hasan Selcuk Ozger, Resul Karakus, Umit Emin Bagriacik, Aysegul Yucel Atak, Esin Senol.

**Project administration:** Esin Senol.

**Writing – original draft:** Hasan Selcuk Ozger, Resul Karakus.

**Writing – review & editing:** Umit Emin Bagriacik, Melda Turkoglu, Gonca Erbas, Aysegul Yucel Atak, Esin Senol.

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
