## [Decision Letter · Decision Letter 0]

1 Jul 2021

PONE-D-21-17711

Serial measurement of cytokines strongly predict COVID-19 outcome

PLOS ONE

Dear Dr. Ozger,

Thank you for submitting your manuscript to PLOS ONE. After careful consideration, we feel that it has merit but does not fully meet PLOS ONE’s publication criteria as it currently stands. Therefore, we invite you to submit a revised version of the manuscript that addresses the points raised during the review process.

ACADEMIC EDITOR: Please revise according to reviewers comments 

We look forward to receiving your revised manuscript.

Kind regards,

Prasenjit Mitra, MD, MRSB, MIScT, FLS, FACSc, FAACC

Academic Editor

PLOS ONE

Journal Requirements:

2. Please provide additional details regarding participant consent. In the ethics statement in the Methods and online submission information, please ensure that you have specified:

 - whether consent was obtained

 - whether consent was informed

 - what type of consent you obtained (for instance, written or verbal, and if verbal, how it was documented and witnessed).

 - if your study included minors, state whether you obtained consent from parents or guardians.

 - if the need for consent was waived by the ethics committee, please include this information

"The funders had no role in study design, data collection and analysis, decision to

publish, or preparation of the manuscript."

"This study was funded by Gazi University Scientific Research Projects Unit."

Reviewers' comments:

Reviewer's Responses to Questions

**Comments to the Author**

1. Is the manuscript technically sound, and do the data support the conclusions?

Reviewer #1: Partly

2. Has the statistical analysis been performed appropriately and rigorously? 

Reviewer #1: No

3. Have the authors made all data underlying the findings in their manuscript fully available?

Reviewer #1: Yes

4. Is the manuscript presented in an intelligible fashion and written in standard English?

Reviewer #1: Yes

5. Review Comments to the Author

Reviewer #1: The authors evaluated the predictive value of serially measured cytokines for 37 hospitalized COVID-19 cases and concluded that serially measured cytokines yield a better prediction for outcome of COVID-19 patients. Although the study appears interesting, there are a few issues that may have to be clarified/rectified.

1. The authors have stated their aim as to assess the value of change in cytokine levels in predicting COVID-19 mortality. To evaluate the same, ROC at different time points have been compared. However the comparison of ROC at different time points would enable the authors only to compare the predictive power of cytokines on one day compared to another. If the authors want to assess the value of change in cytokine levels, they may have to plot ROC with the change in the values.

The authors may go through the following articles for reference:

a. Yamaguchi, Kakuhiro MD, PhDa; Iwamoto, Hiroshi MD, PhDa,∗; Sakamoto, Shinjiro MD, PhDa; Horimasu, Yasushi MD, PhDa; Masuda, Takeshi MD, PhDa; Miyamoto, Shintaro MD, PhDa; Nakashima, Taku MD, PhDa; Ohshimo, Shinichiro MD, PhDb; Fujitaka, Kazunori MD, PhDa; Hamada, Hironobu MD, PhDc; Kohno, Nobuoki MD, PhDd; Hattori, Noboru MD, PhDa Serial measurements of KL-6 for monitoring activity and recurrence of interstitial pneumonia with anti-aminoacyl-tRNA synthetase antibody, Medicine: December 2018 - Volume 97 - Issue 49 - p e13542 doi: 10.1097/MD.0000000000013542

b. Kamarudin, A.N., Cox, T. & Kolamunnage-Dona, R. Time-dependent ROC curve analysis in medical research: current methods and applications. BMC Med Res Methodol 17, 53 (2017). https://doi.org/10.1186/s12874-017-0332-6

2. As the authors have mentioned in limitations, age and comorbidities are pronounced confounders of the study. Authors may have to explore the possibilities of regression analysis to account for the confounding effects of these factors. As different comorbodtites on its own can cause change in cytokine levels, it is imperative to delineate the role of cytokine in mortality independent of the influence of comorbidities. Further regression tools may help the authors in performing the analysis.

Discussion also should include the possible role of these factors in altering the cytokine levels and leading to mortality.

3. I think Surviors and non- survivors would be a better terminology for the non-mortal and mortal groups.

4. Figures lack clarity. Better resolution figures to be included.

6. PLOS authors have the option to publish the peer review history of their article (what does this mean?). If published, this will include your full peer review and any attached files.

Reviewer #1: No

---

## [Author Response · Author response to Decision Letter 0]

11 Aug 2021

08.03.2021

Thank you for your interest and valuable contribution to our article. We have revised our article according to your suggestions and made possible changes. We are resubmitting the article by making the changes suggested by the reviewers. New changes are marked in yellow in the manuscript.

Best regards,

 Selcuk

Journal requirements

1. Our manuscript re-edited according to PLOS ONE's style requirements.

2. Additional details on participant consent added to the method section.

3. The recommended statement added to the acknowledgment section of our article. The text on funding removed from the manuscript.

4. Ethics other than method section deleted.

Answers to reviewer

1-Thank you very much for your suggestions. Our article is primarily based on the idea that single-measure cytokine analyzes with uncertain sampling time may be misleading due to the dynamic course of cytokines. For this reason, serial evaluations were planned. We think that the analysis on the change values you suggested will contribute to the objectives of the article. Therefore, the current analyzes have been reconsidered by taking into account the changes in cytokinin levels at day 0 and day 3, and necessary changes have been made in the article that we think will meet your recommendations.

2-We agree with you that regression analyzes are necessary to determine the role of cytokines in mortality, regardless of the influence of confounding factors such as, comorbidities, age, etc. However, we think that regression analyzes cannot be performed due to the current sample size especially in the non-survivors group. The number of patients included in the study was limited due to the low number of cases in our country during the study period, the fact that our study was conducted in a single center and central restrictions were imposed on multicenter studies related to COVID-19 in our country. Also, the high cost of financing due to the scope of the study, which aimed to evaluate many cytokines, caused the number of patients included in the study to be limited.This limitation is clearly stated in the limitation section of our article.

3-Based on your suggestions, the study groups were named survivors and non-survivors.

4-The figures in the study were evaluated and created in terms of re-resolution and clarity.

---

## [Decision Letter · Decision Letter 1]

14 Sep 2021

PONE-D-21-17711R1Serial measurement of cytokines strongly predict COVID-19 outcomePLOS ONE

Dear Dr. Ozger,

Thank you for submitting your manuscript to PLOS ONE. After careful consideration, we feel that it has merit but does not fully meet PLOS ONE’s publication criteria as it currently stands. Therefore, we invite you to submit a revised version of the manuscript that addresses the points raised during the review process.

ACADEMIC EDITOR: Please modify the manuscript

We look forward to receiving your revised manuscript.

Kind regards,

Prasenjit Mitra, MD, CBiol, MRSB, MIScT, FLS, FACSc, FAACC

Academic Editor

PLOS ONE

Journal Requirements:

Reviewers' comments:

Reviewer's Responses to Questions

**Comments to the Author**

1. If the authors have adequately addressed your comments raised in a previous round of review and you feel that this manuscript is now acceptable for publication, you may indicate that here to bypass the “Comments to the Author” section, enter your conflict of interest statement in the “Confidential to Editor” section, and submit your "Accept" recommendation.

Reviewer #1: (No Response)

2. Is the manuscript technically sound, and do the data support the conclusions?

Reviewer #1: Yes

3. Has the statistical analysis been performed appropriately and rigorously? 

Reviewer #1: Yes

4. Have the authors made all data underlying the findings in their manuscript fully available?

Reviewer #1: Yes

5. Is the manuscript presented in an intelligible fashion and written in standard English?

Reviewer #1: Yes

6. Review Comments to the Author

Reviewer #1: The reviewers have revised the manuscript adequately after addressing the comments and the efforts have to be appreciated.

However, the following points may be noted:

1. For IL-1RA , the values are mentioned as 48 (7.1 to 9.1). Please do check once for typographical errors as median is less than the 75th percentile.

2. For PDGF-AA, the values are mentioned as 4857 ± 2411. From preliminary glance, the distribution appears to be normal. Please verify if the representation as mean ± SD is correct in this context or not.

3. In first line of discussion, it may have to be specifically mentioned that IL-6, IL-7, IL-10, IL-15, IL-27, IP-10, MCP-1, and GCSF cytokines on DAY OF ADMISSION could be used for prediction of mortality in COVID-19 patients.

4. The Figure 1 is not at all legible and is not acceptable in the current state . Figure 1 have to be redrawn.

7. PLOS authors have the option to publish the peer review history of their article (what does this mean?). If published, this will include your full peer review and any attached files.

Reviewer #1: No

---

## [Author Response · Author response to Decision Letter 1]

17 Sep 2021

09.015.2021

Thank you for your interest and valuable contribution to our revised article. We have revised our article according to your suggestions and made possible changes. We are resubmitting the article by making the changes suggested by the reviewers. New changes are marked in yellow in the manuscript.

Best regards,

 Selcuk

Journal requirements

1.We've reviewed our reference list to make sure it's complete and accurate.

References 23 and 24 ( [24] Costela-Ruiz VJ, Illescas-Montes R, Puerta-Puerta JM, Ruiz C, Melguizo-Rodriguez L. SARS-CoV-2 infection: The role of cytokines in COVID-19 disease. Cytokine Growth Factor Rev. 2020;54:62-75., [25] Petrey AC, Qeadan F, Middleton EA, Pinchuk IV, Campbell RA, Beswick EJ. Cytokine release syndrome in COVID-19: Innate immune, vascular, and platelet pathogenic factors differ in severity of disease and sex. J Leukoc Biol. 2021;109:55-66.) in the first manuscript were omitted from the first revision due to the reorganization of the discussion section. In the current revision, the references were checked and no change was made to the references.

Answers to reviewer

1-Thank you very much for your suggestions. The values for IL-1RA in the non-survivor group were corrected to 48 (7.1 - 90.1).

2-You are right, the values for PDGF-AA show normal distribution. For this reason, instead of the median (IQR, 25-75%), it was preferred to presentation as mean ± SD. The error in this presentation was corrected and PDGF-AA values (5011 ± 2884 for non-survivors and 4283 ± 1979 for survivors) were shown.

3-'On the day of admission' was added in the first line of the discussion, according to your suggestion.

4- In our study, the GraphPad (version 9) program was used to create the graphs. Because Figure 1 is a multi-panel graph and many cytokine levels were compared, the desired image resolution could not be achieved. Figure 1 was revised based on your suggestions and your PLOS ONE figure preparation requirements. The pdf format of the graph was checked with the PACE PLOS and GIMP programs. We hope that sufficient quality and resolution have been achieved. But if the problem persists, we can rearrange the graphs by splitting (survivors, survivors, etc.).

---

## [Decision Letter · Decision Letter 2]

15 Nov 2021

Serial measurement of cytokines strongly predict COVID-19 outcome

PONE-D-21-17711R2

Dear Dr. Ozger,

We’re pleased to inform you that your manuscript has been judged scientifically suitable for publication and will be formally accepted for publication once it meets all outstanding technical requirements.

Kind regards,

Etsuro Ito

Academic Editor

PLOS ONE

Reviewers' comments:

Reviewer's Responses to Questions

**Comments to the Author**

1. If the authors have adequately addressed your comments raised in a previous round of review and you feel that this manuscript is now acceptable for publication, you may indicate that here to bypass the “Comments to the Author” section, enter your conflict of interest statement in the “Confidential to Editor” section, and submit your "Accept" recommendation.

Reviewer #1: All comments have been addressed

2. Is the manuscript technically sound, and do the data support the conclusions?

Reviewer #1: Yes

3. Has the statistical analysis been performed appropriately and rigorously? 

Reviewer #1: Yes

4. Have the authors made all data underlying the findings in their manuscript fully available?

Reviewer #1: Yes

5. Is the manuscript presented in an intelligible fashion and written in standard English?

Reviewer #1: Yes

6. Review Comments to the Author

Reviewer #1: (No Response)

7. PLOS authors have the option to publish the peer review history of their article (what does this mean?). If published, this will include your full peer review and any attached files.

Reviewer #1: No

---

## [Editor Report · Acceptance letter]

23 Nov 2021

PONE-D-21-17711R2 

Serial measurement of cytokines strongly predict COVID-19 outcome 

Dear Dr. Ozger:

I'm pleased to inform you that your manuscript has been deemed suitable for publication in PLOS ONE. Congratulations! Your manuscript is now with our production department. 

Kind regards, 

on behalf of

Prof. Etsuro Ito 

Academic Editor

PLOS ONE